# Topological Obstructions and How to Avoid Them

**Babak Esmaeili**[*]
Generative AI Group
Eindhoven University of Technology
b.esmaeili@tue.nl

**Robin Walters**[*]
Khoury College of Computer Sciences
Northeastern University
r.walters@northeastern.edu

**Heiko Zimmermann**
Amsterdam Machine Learning Lab
University of Amsterdam
h.zimmermann@uva.nl

**Jan-Willem van de Meent**
Amsterdam Machine Learning Lab
University of Amsterdam
j.w.vandemeent@uva.nl

## Abstract

Incorporating geometric inductive biases into models can aid interpretability and generalization, but encoding to a specific geometric structure can be challenging due to the imposed topological constraints. In this paper, we theoretically and empirically characterize obstructions to training encoders with geometric latent spaces. We show that local optima can arise due to singularities (e.g. self-intersection) or due to an incorrect degree or winding number. We then discuss how normalizing flows can potentially circumvent these obstructions by defining multimodal variational distributions. Inspired by this observation, we propose a new flow-based model that maps data points to multimodal distributions over geometric spaces and empirically evaluate our model on 2 domains. We observe improved stability during training and a higher chance of converging to a homeomorphic encoder.

## 1 Introduction

A well-established idea in machine learning research is that geometric inductive biases can help us learn representations that reflect the underlying structure of a dataset [Bronstein et al., 2021, Higgins et al., 2022]. A key intuition behind this line of research is that such representations make it easier to reason about the similarities of different instances in the dataset, for example by relating inputs using a rotation or translation, which in turn aids interpretability and data-efficiency. Geometric inductive biases have been explored in a wide variety of forms, including models that are defined on hyperbolic or spherical Riemannian manifolds [Lezcano-Casado and Martınez-Rubio, 2019, Ganea et al., 2018], models that are equivariant or invariant with respect to particular symmetry groups [Kondor and Trivedi, 2018, Cohen and Welling, 2016], and work that leverages symmetries to learn disentangled representations that factorize into distinct axes of variation [Higgins et al., 2018].

In this paper, we consider symmetry-based approaches to learning representations in an unsupervised manner by imposing geometric inductive biases on the representation space. In this context, our notion of a representation that "reflects the underlying structure of the data" is a representation that is *homeomorphic* to the true generative factors. We are specifically interested in the optimization challenges that arise when encoding to geometric spaces such as Lie groups. While a common intuition is that an inductive bias that matches the underlying topology of the data will guide a model towards a homeomorphic representation, there are also indications that certain inductive biases can make a model more difficult to train in practice [Park et al., 2022, Falorsi et al., 2018, Batson et al., 2021], which limits their practical utility.

---

[*]Equal contribution

37th Conference on Neural Information Processing Systems (NeurIPS 2023).

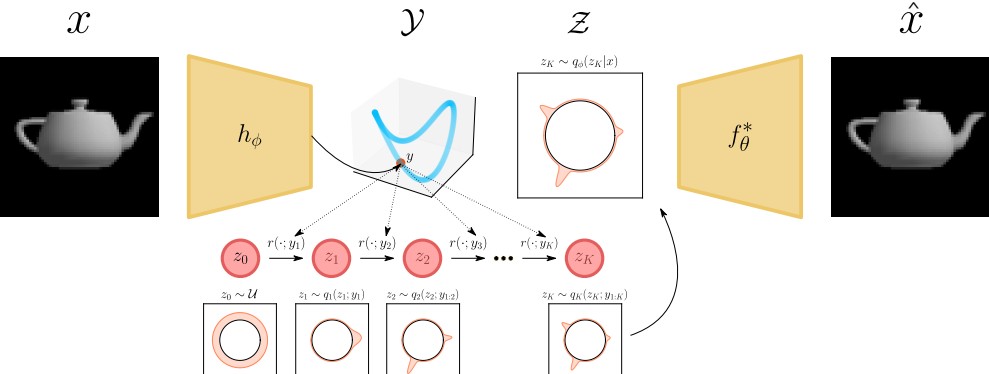

Figure 1: A GF-VAE consists of an encoder network $h_\phi$ that maps data $x$ to an intermediate parameter space $\mathcal{Y}$. The encoded vector $y$ is split into $K$ parts, where each sub-vector $y_k$ corresponds to the parameters of a normalizing flow at layer $k$. We can sample from the variational distribution $q_\phi$ by first sampling from a uniform prior defined on the Lie group ($\mathrm{SO}(2)$ in this case), followed by a sequence of $K$ bijective transformations $r(\cdot; y_k)$. The output of the normalizing flows, denoted as $z_K$, is then passed to a decoder $f_\theta^*$ that maps from the Lie group $\mathcal{Z}$ to the data space $\mathcal{X}$.

To understand why encoding to geometric structures can give rise to optimization challenges, we formalize topological defects that can occur in a randomly initialized encoder, such as discrepancies in the winding number or crossing number relative to those in a homeomorphic encoder. We show that these topological defects will be preserved under continuous optimization, which suggests that escaping these local optima relies on the discrete jumps that are employed during optimization.

To circumvent these obstacles to optimization, we propose Group-Flow Variational Autoencoders (GF-VAEs), which leverage normalizing flows to model complex multimodal distributions on Riemannian manifolds (Figure 1). We show that if we define the mode of the variational distribution to be the representation, normalizing flows can circumvent some of the challenges associated with local optima due to their multimodal nature. Experiments demonstrate that GF-VAEs can escape local optima during the early stages of training, resulting in more reliable convergence to a homeomorphic mapping and a greater degree of continuity after training.

We summarize the main contributions of this paper as follows:

- We characterize topological defects that can arise in encoders that map onto spaces with geometric structure. We show that some obstructions that arise from topological defects cannot be resolved using continuous optimization.

- We define evaluation criteria based on the winding number, crossing number, and continuity to measure topological defects and homeomorphism violations in the encoder.

- We propose GF-VAEs, a new VAE-based model that that employs normalizing flows to define complex distribution on Lie groups. We empirically show that GF-VAEs are able to aid in circumventing identified optimization obstructions.

## 2    Problem Statement

**Homeomorphic Encoders.**    Our goal is to learn representations in domains where we have prior knowledge of the geometric structure, specifically structure in the form of a symmetry group that can be associated with the input data. We assume that data lies on a low-dimensional manifold $\mathcal{M}$ that is embedded in a higher-dimensional space $\mathcal{X} := \mathbb{R}^n$ via a mapping $g_{\mathcal{X}} : \mathcal{M} \to \mathcal{X}$. This is commonly known as the *manifold hypothesis* [Bengio et al., 2013]. We denote the image of the mapping by $\mathcal{M}_x := g_{\mathcal{X}}(\mathcal{M}) \subseteq \mathcal{X}$. Then $g_{\mathcal{X}}$ is a homeomorphism, or topological isomorphism, onto its image $g : \mathcal{M} \xrightarrow{\sim} \mathcal{M}_x$. That is, $g$ is continuous, bijective, and has a continuous inverse.

We wish to learn an encoder $f_\phi : \mathcal{X} \to \mathcal{Z}$ such that its restriction $f_\phi|_{\mathcal{M}_x} : \mathcal{M}_x \to \mathcal{Z}$ is a homeomorphism. Following Falorsi et al. [2018], we will define this mapping in terms of a network $h_\phi : \mathcal{X} \to \mathcal{Y}$

that maps to an intermediate space $\mathcal{Y} := \mathbb{R}^d$, followed by a known projection $\pi : \mathcal{Y} \to \mathcal{Z}$,

$$f_\phi \colon \mathcal{X} \xrightarrow{h_\phi} \mathcal{Y} \xrightarrow{\pi} \mathcal{Z}. \tag{1}$$

**Lie Groups.** Our work focuses the specific case where the manifolds $\mathcal{M}$ and $\mathcal{Z}$ are Lie groups. Any set of symmetries may be formally described by a *group* $G$, which is a set of invertible transformations which may be composed using a binary operation $\cdot : G \times G \to G$. A *Lie group* is a group that is also a differentiable manifold. Lie groups describe continuous symmetries such as rotations and translations, and are therefore a natural mathematical setting for any system with spatio-temporal symmetries.

Reasoning about Lie groups is difficult due to their non-flat structure. *Lie algebras* provide a way to study Lie groups by considering the tangent space $\mathfrak{g}$ of the manifold at the identity element. The exponential map $\exp : \mathfrak{g} \to G$ maps points the Lie algebra to points on the group manifold as such. A vector $v \in \mathfrak{g}$ defines a vector field on $G$ using the group action to transport $v$ around $G$. Then $\exp(v)$ is defined to be the point reached by flowing along this vector field for unit time. For more details on Lie groups, we refer the readers to [Hall and Hall, 2013].

**Variational Autoencoders on Lie Groups.** In this paper, we primarily focus on variational autoencoder (VAE) [Kingma and Welling, 2014, Rezende et al., 2014] as a means to learn a homeomorphic embedding. In this setting, we define a generative model by composing a uniform prior $p(z) = \mathcal{U}(z)$ on $\mathcal{Z}$ with a likelihood model $p_\theta(x|z) = \mathcal{N}(x; f_\theta^*(z), \sigma_x^2 I_n)$ that is defined in terms of a decoder network $f_\theta^* : \mathcal{Z} \to \mathcal{X}$. We use the encoder $f_\phi$ to define a variational distribution $q_\phi(z|x)$, whose design we discuss below and train the encoder and decoder jointly by optimizing the variational lower bound,

$$\mathcal{L}_{\phi,\theta}^{\text{VAE}}(x) = \mathbb{E}_{q_\phi(x|z)} \left[\log p_\theta(x|z)\right] - D_{\text{KL}} \left[q_\phi(z|x) \| p(z)\right] \leq \log p_\theta(x). \tag{2}$$

Defining a variational distribution on a manifold is generally not straightforward as it requires finding an expression of the density on the manifold or keeping track of the change of volume when projecting to the manifold from the tangent space. Falorsi et al. [2019] define a reparameterized construction for sampling from a Gaussian distribution on $\mathcal{Z}$ by first sampling $\epsilon \sim \mathcal{N}(0, I_p)$ from the $p$-dimensional Lie algebra associated with $\mathcal{Z}$, then rescaling $\epsilon$ by way of element-wise multiplication $\sigma_\phi(x) \odot \epsilon$ using a network $\sigma_\phi : \mathcal{X} \to \mathbb{R}^p$, and computing the corresponding element $z_\epsilon$ on the group manifold using the exponential map. By left multiplying this randomly sampled $z_\epsilon$ by the group element $f_\phi(x) \in \mathcal{M}$ to *move* the mode of the final distribution to its intended location, we obtain the construction

$$\epsilon \sim \mathcal{N}(\cdot; 0, I_p), \quad z_\epsilon = \exp(\sigma_\phi(x) \odot \epsilon), \quad z = f_\phi(x) \cdot z_\epsilon, \quad p(z_\epsilon) = p(\epsilon) \left| \det \frac{\partial \exp(\epsilon)}{\partial \epsilon} \right|^{-1}. \tag{3}$$

We account for the possible change in volume using the change of variable formula. Note that composing group elements $f_\phi(x) \cdot z_\epsilon$ does not change the density of the resulting point on the manifold because the *Haar* measure, a standard choice for Lie groups, is left invariant.

**Running Example.** As a concrete running example, which we will use throughout this paper, we will consider data on the circle $\mathcal{M} = \text{SO}(2)$ that is embedded into a space of images $\mathcal{X}$. The subset $\mathcal{M}_x$ of images generated by a function $g \colon \text{SO}(2) \to \mathcal{M}_x$ corresponds to images of an object that is subject to a one-dimensional rotation. We will consider the case of in-plane rotations of images, as well images of a 3-dimensional object that is rotated around a single axis.

The special orthogonal Lie group $\text{SO}(2)$ is defined as

$$\text{SO}(2) := \left\{ z \mid z \in \text{GL}(2), z^T z = I, \det(z) = 1 \right\} = \left\{ A(y) := \begin{bmatrix} y_1 & -y_2 \\ y_2 & y_1 \end{bmatrix} \mid y \in \mathbb{R}^2, \|y\| = 1 \right\},$$

where $\text{GL}(2)$ is the general linear group, which is the group of invertible $2 \times 2$ matrices under matrix multiplication. The set of images $\mathcal{M}_x$ lives on a manifold that is homeomorphic to the rotation group $\text{SO}(2)$, embedded into the space of images by $g$. Because we assume we know the underlying manifold $\mathcal{M}$, we can design $\mathcal{Z}$ to have the same structure. When $\mathcal{M} = \text{SO}(2)$, we can use $\mathcal{Y} := \mathbb{R}^2$. The function $h_\phi$ in Equation 1 is then an ordinary neural network, and the projection to $\mathcal{Z}$ is simply the projection onto the unit circle $: \mathbb{R}^2 \to \text{SO}(2) := y \mapsto A(y/\|y\|)$.

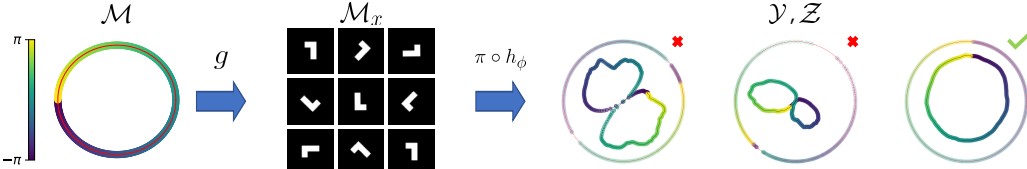

Figure 2: Example of topological defects in learned encoders for a VAE with data on $\mathcal{X} = \mathbb{R}^{32x32}$ in the form of rotations L-shaped tetrominoes and $\mathcal{Z} = \mathrm{SO}(2)$ (the unit circle). Our goal is to learn an encoder $f_\phi$ that defines a homeomorphism (a continuous bijection with continous inverse) between the manifold of images of L-shaped tetrominoes $\mathcal{M}_x \subset \mathcal{X}$ and that of the latent space $\mathcal{Z} = \mathrm{SO}(2)$. The encoder $f_\phi = \pi \circ h_\phi$ is defined by composing a network $h_\phi : \mathcal{X} \to \mathcal{Y}$ with a projection $\pi : \mathcal{Y} \to \mathcal{Z}$. On the right, we show the intermediate space $\mathcal{Y}_x = h_\phi(\mathcal{M}_x)$ and its projection $\mathcal{Z}_x = \pi(\mathcal{Y}_x)$ for 3 random seeds after convergence. Colours indicate the angle associated with each data point on the manifold $\mathcal{M}$. Optimization obstructions can arise when the network $h_\phi$ maps data onto a trajectory $\mathcal{Y}_x$ that exhibits topological defects, such as the crossing in a "figure 8" shape, which gives rise to discontinuities in the projection $\mathcal{Z}_x$ onto the latent space.

## 3   Optimization Obstructions

In practice, training homeomorphic encoders can give rise to optimization challenges. To develop intuition for these challenges, we will consider the running example $\mathcal{M} = \mathrm{SO}(2)$ with data in the form of in-plane rotated images on $\mathcal{X} = \mathbb{R}^{32 \times 32}$ (Figure 2). We train a VAE in which the network $h_\phi$ is a standard convolutional network which is paired with a deconvolutional decoder $f^* : \mathrm{SO}(2) \to \mathbb{R}^{32 \times 32}$ (see Appendix A). Figure 2 shows encodings in the intermediate space $\mathcal{Y}$ after training with ELBO with 3 random seeds. For the first two seeds, we see that $h_\phi(\mathcal{M}_x)$ crosses over itself, resulting in an figure "8" shape. When projected onto $\mathcal{Z} := \mathrm{SO}(2)$, this results in discontinuities at the crossover points. The second seed in addition also exhibits a sparse region in the intermediate space, leading to a gap in the $\mathrm{SO}(2)$ projection. Only the third initialization has converged to the correct topology. These local optima are not unique to this example; obstructions have also been encountered in Falorsi et al. [2018] when trying to learn 3D orientations of a rotating multi-color cube from 2D images using a homeomorphic VAE. Park et al. [2022] show that the homeomorphic VAE cannot generalize well to other shapes and tends to learn a degenerate embedding to a small part of $\mathrm{SO}(3)$.

The main observation that we make in this paper is that imposing a geometric structure on the latent space can introduce topological obstructions *during optimization*. This insight is distinct from the homological obstructions identified by de Haan and Falorsi [2018], who describe the obstructions that emerge from the choice of parameterization on the Lie group. We will refer to the topological defects that we identify in this paper as "optimization obstructions". The problem that we identify here is that randomly-initialized layers have a high probability of exhibiting topological defects (degree, crossing, etc.) that cannot be resolved under continuous optimization using gradient flow. Removal of these topological defects is thus only possible by relying on the jumps coming from performing SGD with a large enough learning rate. This implies that while escaping such local minima is possible, it is difficult and may require many epochs to do so, dramatically slowing training and undercutting the advantages of learning a homeomorphic representation.

We now discuss several specific optimization obstructions. We focus on the case where $\mathcal{M}$ is the Lie group $\mathrm{SO}(2)$, with the same example setting described in Section 2. All the optimization obstructions we consider occur in this case and in the case of higher-dimensional manifolds $\mathcal{M}$ as well, but are simpler to describe for $S^1$. See Appendix B for proofs.

### 3.1   Figure Eight Local Minima

To more precisely describe obstructions that might arise during optimization, we consider continuous-time training along a gradient flow. We denote the weights of the initialized encoder as $\phi(0)$ and the trained weights as $\phi(1)$. We consider the idealized setting in which $\phi(t)$ is a continuous function of $t$.

Empirically, either at initialization or after some training, we often observe a "figure 8" pattern in $\mathcal{Y}$ of roughly the form $(h^\infty \circ g)(\theta) = (\cos \theta \sin \theta, \sin \theta)^\top$. In such a case, the embedding into $\mathcal{Z}$ contains two disjoint pieces $[-3\pi/4, -\pi/4] \cup [\pi/4, 3\pi/4]$. One half of the circle is mapped to one piece at

the top of the circle and the other half of the circle is mapped to the other disjoint piece at the bottom of circle. Such a mapping takes advantage of the singularity at $(0,0)$[2] into order to embed $S^1$ in two continuous pieces. The resulting mapping is *nearly* bijective, failing to reconstruct on only a small region near the two discontinuities. It is also mostly continuous, having only two discontinuities.

Once this local minimum is obtained, it is very difficult to move out of it using gradient descent. It is unlikely for the two pieces to join together and become a homeomorpic embedding since this would require passing one disjoint segment through the other and reversing its orientation which would violate bijectivity and increase the reconstruction loss.

We make this observation precise by noting that continuous optimization preserves the ordering of points on the circle. Let $(z_1, z_2, z_3, z_4)$ denote the four end points of the two disjoint intervals of $f_{\phi(0)}$. The ordering of these points is only defined up to cyclic permutations $\mod C_4$. Proposition 3.1 states that continuous optimization must preserve ordering $\mod C_4$. Figure 3 illustrates the proof.

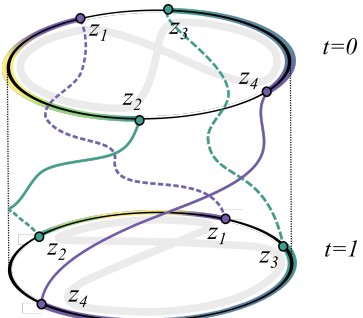

**Proposition 3.1** *Assume that $f_{\phi(t)}$ undergoes continuous optimization. Assume that $f_{\phi(t)} \circ g$ is injective for all $t$. The cyclic ordering induced on $k$ points by $f_{\phi(0)}$ is equal to $f_{\phi(1)}$. Thus a figure 8 embedding, which corresponds to cyclic order $(z_1, z_2, z_4, z_3) \mod C_4$, cannot be transformed to a homeomorphic embedding, which has cyclic order $(z_1, z_2, z_3, z_4) \mod C_4$ or $(z_4, z_3, z_2, z_1) \mod C_4$.*

Figure 3: The figure 8 pattern in $\mathcal{Y}$ (grey) maps to two disconnected components in $\mathcal{Z}$. The cyclic order of these 4 endpoints is preserved by homotopy. Following the parameterization of the data manifold the cyclic order is $(z_1, z_2, z_4, z_3)$, which is distinct from the cyclic order of a homeomorphic embedding, either $(z_1, z_2, z_3, z_4)$ or $(z_4, z_3, z_2, z_1)$.

In other words, transition from a figure 8 embedding to a homeomorphic embedding is impossible without violating continuity during optimization. This indicates that escaping a figure 8 local optimum during training would need to rely on discrete jumps and likely the stochasticity of the gradient estimate.

### 3.2 Degree Obstructions

A second class of topological defects that can arise are encodings with a discrepancy in the winding number. We can compute the degree, or winding number, of a map $\psi : S^1 \to S^1$ around the origin by summing up the differentials along its path on the sphere,

$$\omega(\psi) = \frac{1}{2\pi} \int_{S^1} d\psi(\theta).$$

This concept can be extended to arbitrary connected oriented manifolds, where it is usually referred to as the degree of a mapping. Intuitively, it describes the number of times that the *domain manifold* wraps around the *co-domain manifold*.

If the embedded image $h_{\phi(t)}(\mathcal{M}_x)$ does not contain the origin, then the mapping factors through $\mathbb{R}^2 \setminus \{(0,0)\}$ and is consequently continuous. Therefore, a continuous path in $\phi$ yields a continuous path in $\mathcal{Z}$. If $f_\phi|_{\mathcal{M}_x}$ is continuous, then the function $\psi_\phi := \pi \circ h_\phi \circ g : S^1 \to S^1$ has a well-defined degree, also known as winding number $w(\psi_\phi) \in \mathbb{Z}$. In order for $f_\phi|_{\mathcal{M}_x}$ to be a homeomorphism, the winding number must be $w(\psi_\phi) \in \{-1, 1\}$. Under random initialization, however, the initial network may have winding number equal to any integer. Assuming continuous optimization and continuous embeddings, then $h_{\phi(t)}(x)$ is a continuous function of both $x, t$. We assume that $h_{\phi(t)}(x) \neq (0,0)$ for any $t, x$ and thus winding number is defined for any time. The following proposition thus holds.

**Proposition 3.2** *Winding numbers of the initialized and final model are equal $w(\psi_{\phi(0)}) = w(\psi_{\phi(1)})$.*

In practice, neither the continuous optimization assumption nor the avoidance of the origin holds. Rather $h_\phi$ is updated by SGD in discrete jumps and $h_\phi \circ g$ may map to the origin. Thus, empirically, we do see that the winding number may change during training. However, if the initialization avoids

---

[2]In practice, we never exactly cross $(0,0)$ but only pass by it with a small distance.

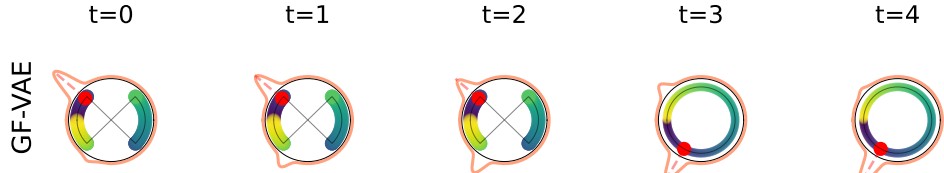

Figure 4:   The initialized encoder (t=0) may contain defects as described in Section 3, such as a figure 8 pattern. With a standard VAE, where the learned representation (red point) is the mean of a conditional gaussian, these defects are unlikely to be resolved during optimization, as it would require passing through intermediate parameters (and hence representations) with higher reconstruction loss. Using a multimodal variational distibution, the parameters and corresponding representations are less tightly coupled, in the sense that a continuous change in the parameter space can result in a discontinues change in representation (i.e. the mode of the distribution) without passing through *high-loss* areas of the parameters space.

the origin, then due to the tendency of the magnitude of the unnormalized embeddings $h_\phi(x)$ to grow during optimization (see Section 3.3), the winding number changing becomes more unlikely. This means that defects in the winding number pose significant obstruction to learning homeomorphic embeddings. The winding number is also the primary optimization obstruction which makes it impractical to remove the hard projection $\pi$. If instead we decode directly from $y \in \mathcal{Y}$ but push embeddings to the unit circle using the loss $|\|y\| - 1|$, then it is far more likely we converge to discontinuous embeddings with the incorrect winding number (Appendix D).

### 3.3   Magnitude Growth in $\mathcal{Y}$

Empirically, we observe the values of the embeddings in $\mathcal{Y} := \mathbb{R}^2$ continually grow during training. This phenomenon makes it more difficult for the embedded data manifold $h_\phi(\mathcal{M}_x)$ to cross the origin and for the winding number to change. We give a theoretical explanation for this behavior.

Consider what would happen if the embedding $y$ were updated directly based on the gradient of the loss $\nabla_y \mathcal{L}$ with respect to $y$. We assume the loss depends only on $z = y/\|y\|$, and so has level sets which are unions of radial rays from the origin. The gradient $\nabla_y \mathcal{L}$ must then be tangent to a circle about the origin. That is, for $y = (a, b)$, the gradient $\nabla_y \mathcal{L} = (\pm b, \mp a)$. Under gradient flow, the evolution of $y$ in time $y_t$ would thus flow along circles of fixed radius and so $\|y_0\| = \|y_t\|$. Under gradient descent, however, due to the convexity of the flow lines, which are circular, the embeddings $y$ will tend to grow in magnitude. For $\eta \in \mathbb{R}_{>0}$, we compute

$$\|y - \eta \nabla_y \mathcal{L}\|^2 = (a \mp \eta b)^2 + (b \pm \eta a)^2 = (a^2 + b^2)(1 + \eta^2) > \|y\|^2.$$

In practice, however, we do not update $y$ based on $\nabla_y \mathcal{L}$ but rather based on the gradient with respect to model parameters $\phi$. Let $F: \Phi \to \mathcal{Y}$ be the map from model parameters $\phi$ to $y$ given fixed input data $x$. Then the actual update to $y$ is $\tilde{\nabla}_y \mathcal{L} = dF^T \circ \nabla_\phi \mathcal{L}$ where $dF$ is the total derivative or Jacobian of the map $F$. Since $\nabla_\phi \mathcal{L} = (dF) \nabla_y \mathcal{L}$ we have $\tilde{\nabla}_y \mathcal{L} = dF^T dF \nabla_y \mathcal{L}$. The angle between $\tilde{\nabla}_y \mathcal{L}$ and $\nabla_y \mathcal{L}$ is bounded by some $\theta$ a quantity depending on the eigenvalues of the operator $dF$. Given that $\nabla_y \mathcal{L}$ is tangential to the circle, assuming for simplicity $\tilde{\nabla}_y \mathcal{L}$ has constant length $L$ and uniform distribution $[-\theta, \theta]$ in angle to $\nabla_y \mathcal{L}$, the norm of $y$ still grows *in expectation*.

**Proposition 3.3** *Assume a circle of radius $R$. Let $v$ be a random vector at $y$ on the circle of length $L$ with angle to the tangent uniform in $[-\theta, \theta]$. Then*

$$\mathbb{E}[\|y + v\|^2] = L^2 + R^2 > R^2 = \|y\|^2.$$

## 4   GroupFlow-VAE

The topological obstructions to optimization that we identified in Section 3 cannot be resolved under continuous optimization. Moreover, even if we allow for a degree of discontinuity, resolving obstructions like a figure 8 will require violating bijectivity, which implies that the reconstruction loss

must increase to escape this defect. This suggests that a VAE with a reparameterized construction as described in Equation 3 will be susceptible to local optima, which aligns with the empirical observation that homeomorphic VAEs can be difficult to train.

This raises the question of whether we can make VAEs less susceptible to topological obstructions by employing a different parameterization from that of Equation 3. More concretely, we will consider a parameterization that admits multiple modes in the variational distribution $q_\phi(z|x)$, rather than the unimodal construction in Equation 3, and define $f_\phi(x)$ in terms of the mode,

$$f_\phi(x) := \arg\max_z q_\phi(z|x). \tag{4}$$

The intuition behind this approach is illustrated in Figure 4. If the variational distribution contains multiple modes, rather than a single peak centered at $\pi(h_\phi(x))$, then it may becomes possible for small changes to result in non-local changes, such as the reordering of points that is needed to untwist a figure 8, by switching between modes in the variational distribution. Moreover, increasing the number of parameters of $q_\phi(z|x)$ is likely beneficial for escaping some of the topological obstructions as some defects such as self-intersection are less likely to occur in high-dimensional spaces.

Motivated by this intuition, we consider a parameterization of $q_\phi(z|x)$ that employs normalizing flows [Rezende and Mohamed, 2015]. In a normalizing flow, the sample $z$ is defined as a push forward of sequence of smooth bijective transformations, which makes it possible to reshape a simple unimodal distribution into a more complex multimodal distribution. The probability of a sample from the final density can be computed by repeatedly applying the rule for change of variables. Concretely, given a base distribution $p(z_0)$ and a sequences of bijective transformations $r_k : \mathcal{Z} \to \mathcal{Z}$, we obtain a sample $z = z_K$ and probability by first sampling $z_0 \sim p(z_0)$ and defining a sequence of transformations

$$z_k = r_k(z_{k-1}), \qquad \log p(z_k) = \log p(z_{k-1}) - \log \left| \det \frac{\partial\, r_k(z_{k-1})}{\partial\, z_{k-1}} \right| \qquad \text{for } k = 1 \cdots K. \tag{5}$$

Normalizing flows have been used to define distributions on geometric structures such as Lie groups by either defining a flow on the Lie algebra and computing the push-forward density of the exponential map [Falorsi et al., 2019], or designing structure-specific transformations [Rezende et al., 2020].

In the GF-VAE, we will define a construction in which the encoder network returns the parameters of the flow. We show an overview of the architecture in Figure 1. We define a network $h_\phi : \mathcal{X} \to \mathcal{Y} := \mathbb{R}^{K \times l}$ where $K$ and $l$ are the number of flow layers and parameters respectively, a sequence of bijective transformations $\{r(\cdot; y_k)\}_{k=1}^K$ parameterized by $\{y_k\}_{k=1}^K$, and a base distribution distribution $p(z_0)$ which we define as a distribution on the group. The sequence $\{r(\cdot; y_k)\}_{k=1}^K$ is then used to define a new distribution on $\mathcal{Z}$ given an $x$ by transforming the base distribution. This defines a conditional flow that can be trained using a stand lower bound (Eq. 2),

$$z_0 \sim p(z_0), \qquad z_k = r(z_{k-1}; y_k), \qquad \log q(z_k|x) = \log q(z_{k-1}|x) - \log \left| \det \frac{\partial\, r(z_{k-1}; y_k)}{\partial\, z_{k-1}} \right|.$$

The choice of the flow $r$ is very important here as normalizing flows are typically defined on flat spaces. This means that, for a specific manifold $\mathcal{M}$, additional care must be taken when designing them to ensure that they are a diffeomorphism from $\mathcal{Z}$ to itself [Rezende et al., 2020, Durkan et al., 2019, Mathieu and Nickel, 2020]. In this paper, we employ a similar method as Falorsi et al. where we define an affine layer followed by a single layer of spline flow, and a Tanh layer multiplied by $\pi$ as the last layer to push all the probability density between $-\pi$ and $\pi$.

## 5  Experiments

We perform a series of experiments to evaluate the difficulty of learning a homeomorphic encoder. Concretely, we investigate how often we fall into one of the failure cases described in Section 3 in standard VAEs with geometric latent spaces in practice. Subsequently, we examine how well GF-VAE can circumvent these topological obstructions during training.

**Baselines.**   Throughout our experiments, we compare against (1) a standard VAE, and (2) a supervised VAE where in addition to maximizing the ELBO, the encoder is trained to predict the ground truth representations. These two scenarios serve as extremes on the spectrum of guiding the model to the right representation. We also compare against a deterministic autoencoder (AE) which does not

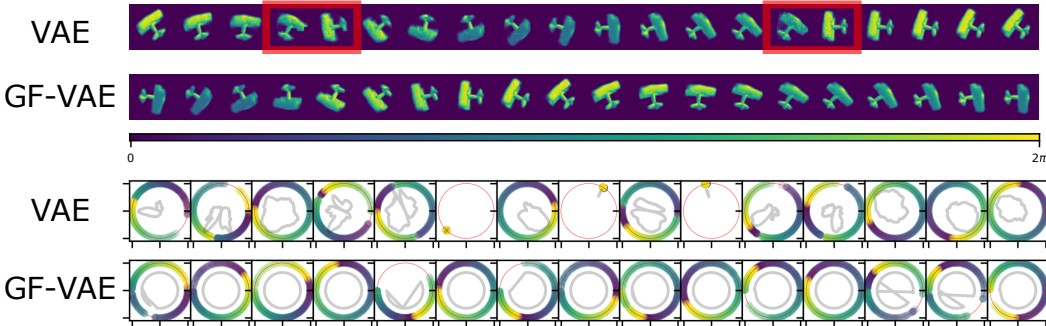

Figure 5: *Top*: Latent traversals in the decoder for a VAE and GF-VAE after training. *Bottom:* The representations in latent space $\mathcal{Z}$ for VAEs and GF-VAEs initialized with 15 different random seeds. For VAEs, the gray line inside the circle show the (scaled-down) $y$-traversals. For GF-VAEs, we it is difficult to visualize $\mathcal{Y}$ space as it is high-dimensional space. Therefore, to inspect for obstructions, we show the traversal of the $z$ vectors instead.

regularize the embedding space. We also tried regularizing the y-space to be close $S_1$ (by penalizing $(\|y\|_2 - 1)^2$) in order to mitigate the optimization problem discussed in Subsection 3.3, which we refer to as "reg-$y$" loss. Finally, we evaluate the $\beta$-VAE objective [Higgins et al., 2017] which increases the regularization on the latent space by upweighting the KL term in Eq 2. All models employ a 4-layer CNN architecture for the encoder and decoder with LeakyReLU activations. For the decoder, we also experiment with *action-decoder* proposed by Falorsi et al., which we found helpful for learning a homeomorphic mapping. The action-decoder uses a special first layer where the group action is applied to a set of learned Fourier coefficients rather than directly being passed as input to the architecture. For further details regarding our experiments, please refer to Appendix A.

**Evaluation.** We evaluate all models based on two criteria: (1) Has the encoder learned a homeomorphic mapping? and (2) Has the decoder learned a good model of the data? Assessing whether a learned mapping is homeomorphic is challenging. To verify homeomorphism, we follow the evaluation proposed in Falorsi et al. [2018] by examining whether the encoder yields a continuous path when interpolating in the data manifold from $-\pi$ to $\pi$. Details on evaluating continuity are provided in Appendix C. To determine how often the models encounter the topological obstructions described in Section 3, we also report crossing and winding numbers in Appendix E. A crossing number grater than 0 implies a "figure 8" obstruction, and a winding number that is not equal to $1$ or $-1$ implies winding number obstruction. Lastly, we measure the log-likelihood to assess how well each model approximates the data manifold. If the model has diverged during training due to posterior collapse, we report it as a non-homeomorphic mapping and ignore its continuity score in the average.

## 5.1 Images: SO(2)

In our first experiment, we train on images of an L-shaped tetromino [Bozkurt et al., 2021], a teapot, and an airplane [Shilane et al., 2004]. The SO(2) manifold corresponding to each object is made by rotating the image of the object around the center. We report our findings in Tables 1 and 3.

Table 1: Number of learned homeomorphic encoders and their continuity score for different objectives trained with 15 different random seeds.

|  | *L-shaped Tetromino* | | *Teapot* | | *Airplane* | |
|  | # H. | Continuity | # H. | Continuity | # H. | Continuity |
| --- | --- | --- | --- | --- | --- | --- |
| AE | 2/15 | $137.28 \pm 54.15$ | 0/15 | $173.37 \pm 21.99$ | 0/15 | $132.95 \pm 40.61$ |
| VAE ($\beta = 1$) | 0/15 | $117.02 \pm 23.17$ | 6/15 | $14.21 \pm 7.30$ | 0/15 | $86.72 \pm 65.12$ |
| VAE ($\beta = 4$) | 2/15 | $132.45 \pm 58.79$ | **15/15** | **$1.22 \pm 0.05$** | 5/15 | $67.88 \pm 59.20$ |
| VAE + $y$-reg | 1/15 | $152.71 \pm 74.20$ | 1/15 | $14.51 \pm 81.$ | 3/15 | $124.10 \pm 93.53$ |
| GF-VAE ($\beta = 1$) | 5/15 | $50.51 \pm 58.18$ | 7/15 | $21.07 \pm 24.42$ | 0/15 | $63.94 \pm 34.43$ |
| GF-VAE ($\beta = 4$) | 9/15 | $24.04 \pm 29.14$ | 13/15 | $8.16 \pm 17.89$ | 7/15 | **$27.70 \pm 29.23$** |
| Action-GF-VAE ($\beta = 4$) | **10/15** | **$19.35 \pm 23.37$** | 13/15 | $3.14 \pm 1.74$ | **9/15** | $33.26 \pm 40.03$ |
| Sup-VAE | **15/15** | **$5.74 \pm 0.49$** | 13/15 | $5.67 \pm 0.34$ | 0/15 | $96.78 \pm 38.$ |

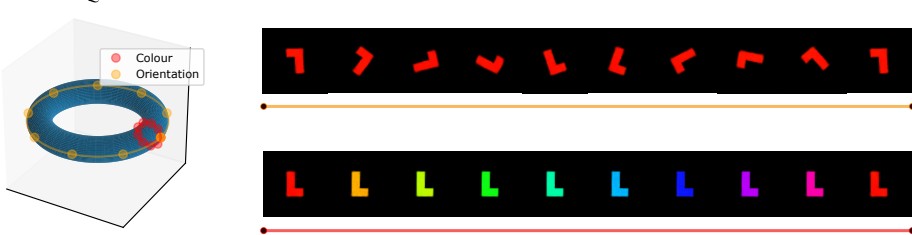

Figure 6: Latent traversals in the decoder of a GF-VAE trained on Tetrominoes with a torus latent spaces. In each traversal, we keep the value for one Lie group fixed and do a geodesic interpolation across the other group. The traversals in the latent space highlighted are by red and orange.

We observe that even though the types of obstructions vary across images, both VAE and AE in general fail to learn a homeomorphic encoder. The GF-VAE objective improves performance noticeably across both metrics. We also did not find $y$ regularization to be very helpful as it mainly stabilized training towards whatever mapping that was achieved at the early stages of training. We observed the main failure case for most of the non-homeomorphic encoders was due to discontinuity emerging from figure-eight obstruction. Moreover, looking at the learning curves in Figure 9, we observe that the winding number is susceptible to change during training which suggests that the continuity assumption is of importance in proposition 3.2. GF-VAE resolves both issues, which allows us to interpolate nicely in the $\mathcal{Z}$-space (Figure 5). What is very surprising is that in the case of airplanes, we see that even supervised objective fails to overcome these optimization obstructions, while a GF-VAE is able to achieve this at a much better rate.

We also observe that increasing the $\beta$ value for the KL term generally helps. This is perhaps unsurprising given that a high $\beta$ value encourages the latent space to cover the prior and therefore discourages the winding numbers from being 0 and improves the performance on the continuity metric. In the case of the teapots, we in fact observe that increasing $\beta$ is sufficient to learn a homeomorphic encoder. However, GF-VAE still generally scores better in terms of continuity and winding numbers.

## 5.2 Images: Torus

In this experiment, we consider a ring torus as the latent space, which is homeomorphic to the Lie group $SO(2) \times SO(2)$. We create a dataset homeomorphic to this group by independently rotating the L-shaped tetromino in both orientation and color. All models can be extended to Tori by simply duplicating the latent space and learning multiple encodings to $SO(2)$ in parallel. This is done by defining a factorized density $q_\phi(z^{(1)}, z^{(2)}|x) = q_\phi(z^{(1)}|x)q_\phi(z^{(2)}|x)$, where each distribution $q_\phi(z^{(i)}|x)$ defines a distribution on a $S^1$. Ideally, we want one subgroup to correspond to colour and the other to orientation. In this setting, we evaluate homeomorphism by picking 10 different values in either colour or orientation and measuring the continuity of the encoded path when interpolating in the data manifold on the other attribute. We identify the encoder as homeomorphic if the average continuity score of all 10 paths is below a certain threshold (see Appendix C). Unsurprisingly, we found that learning a homeomorphic mapping in tori is more challenging compared to circles. Out of 15 runs, the VAE models learn a homeomorphic mapping 1 time, while a GF-VAE manages to learn a homeomorphic mapping 7 times. To be able to align each Lie group with the corresponding attribute, we also used the weakly supervised proposed by Locatello et al., where the model receives a sequence of images in which only the angle or the colour is changing. We show the latent traversal in the decoder model for one of the successful runs of GF-VAE in Figure 6. As we can see, the model has successfully managed to disentangle colour and orientation in the latent space.

## 6 Related Work

**Learning Geometric Representations.** There has been a large amount of work concerned with learning representations from data with geometric structure in the unsupervised or weakly supervised setting. One line of work has focused on the use of geometric spaces such as lie groups in latent space [Davidson et al., 2018, Falorsi et al., 2018, Perez Rey et al., 2020, Vadgama et al., 2022]. In [Miao et al.], it was argued that that geometric inductive biases in VAEs should be Incorporated via a deterministic mapping rather than the prior which is consistent with our work. As we show in this

work, naively incorporating a geometric bias leads to topological obstructions. Another line of work has focused on inferring the latent structure by exploring the local connectivity information [Moor et al., 2020, Chen et al., 2021, Lee et al., 2021, Pfau et al., 2020].

**Learning Disentangled Representations.** Topological group structure can be used to define a notion of disentanglement that is based on equivariant properties under group transformations [Higgins et al., 2018]. There exists a body of work that aims to learn both disentangled and equivariant group representations. One set of methods relies on agent actions to predict the group element [Caselles-Dupré et al., 2019, Quessard et al., 2020]. An adjacent line of work is to regularize encoders to be equivariant with respect to the group action, using triplets of the form $(x_t, m_t, x_{t+1})$ or longer sequences [Guo et al., 2019, Dupont et al., 2020, Tonnaer et al., 2022]. Some approaches focus on learning symmetry-based disentangled representations in fully unsupervised settings or by enforcing commutativity in the latent Lie group [Yang et al., 2022, Zhu et al., 2021]. Another related area disentangles *class* and *pose* in the latent space [Marchetti et al., 2022, Winter et al., 2022]. Our work does not focus on disentanglement, but the topological obstructions that we describe are relevant to this domain.

**Topological Obstructions in Learning.** The topological obstructions that we consider in this work are distinct from the homological obstructions that have been characterized in prior work [de Haan and Falorsi, 2018, Batson et al., 2021, Falorsi et al., 2018, Perez Rey et al., 2020]. Theorem 1 by de Haan and Falorsi [2018] defines homological obstructions as follows: For any latent space $\mathcal{Z}$ with non-trivial topology, it is possible to learn an encoder $f_\phi$ that is continuous when restricted to $\mathcal{M}_x \subset \mathcal{X}$, but this encoder must be discontinuous on the full space $\mathcal{X}$. For this reason, Falorsi et al. [2018] and others [Xu and Durrett, 2018, Meng et al., 2019] use the two-part encoder from equation 1, inserting discontinuous layers $\pi$ when mapping to circles, spheres, $\mathrm{SO}(n)$, or other manifolds. This explicit discontinuity circumvents the homological obstruction without forcing the linear layers of the network to approximate discontinuities using large weights, which we and others find leads to instability during training and inferior reconstructions (Section 5).

**Normalizing Flows on Manifolds.** In recent years, there has been a surge of interest in extending normalizing flows, originally formulated for Euclidean spaces, to Riemannian manifolds [Rezende et al., 2020, Mathieu and Nickel, 2020, Köhler et al., 2021, Durkan et al., 2019]. One approach involves leveraging the Lie group structure of the manifold to define a parametrization of the flow [Rezende et al., 2020], which is also the strategy we adopt in this work. These recent advancements have paved the way for applying normalizing flows to the group $\mathrm{SO}(3)$ in order to learn pose estimation in molecular structures [Köhler et al., 2023] and images [Liu et al., 2023, Murphy et al., 2021]. Our work differs from these efforts in that the primary objective of our method is not to target flexible distributions on Riemannian manifolds, but rather to demonstrate that utilizing a flow as a variational distribution aids *optimization* in learning a homeomorphic embedding.

## 7 Conclusion

In this paper, we investigate obstructions to optimization that can arise when learning encoders for topological spaces. We classify different types of obstructions, provide evidence these are encountered in practice, and give mathematical explanations for how they occur under certain assumptions. We propose GF-VAE, a novel model that employs normalizing flows as variational distributions to help circumvent these issues and show its effectiveness across several datasets when encoding to circles and tori. This work contains several limitations. Firstly, our theoretical analysis is limited by the idealized assumptions necessary to analyze the method using topological tools which do not exactly match those encountered in practice. Secondly, the metrics we define such as winding and crossing numbers are harder to define and compute for higher dimensional manifolds. Future work includes expanding our analysis and techniques to a wider array of Lie groups and non-group manifolds.

## Acknowledgments and Disclosure of Funding

This work was supported by NSF grants #2107256 and #2134178. We also would like to thank Patrick Forré, Sharvaree Vadgama, Marco Federici, and Erik Bekker for helpful discussions.

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

# A  Experimental Details

We train all our models for 150 epochs with a batch-size of 600. For optimization, we use the RAdam optimizer [Liu et al., 2019] with a learning rate of 5e-4. For all image datasets, we use a 4-layer CNN with kernel, stride, and padding of size 4, 2 and 1 respectively followed by a $leakyReLU$ activation (Table 2) for the encoder and a $Sigmoid$ activation for the decoder. The network $\sigma_\epsilon^2$, shares the same architecture with the only difference being the last layer, which is a fully-connected network followed by a $Softplus$ activation which is common in standard VAEs. In all our experiments, we used $K = 1$ for the GF-VAE models as it was sufficient to avoid the optimization obstructions mentioned in the paper. All models were initialized and trained with $15^3$ different random seeds.

Table 2: Architecture of the encoders and decoders employed for all image datasets.

| Encoder |
| --- |
| Input $32 \times 32$ images |
| $4 \times 4$ conv. 32 stride 2, LeakyReLU. |
| $4 \times 4$ conv. 32 stride 2, LeakyReLU. |
| $4 \times 4$ conv. 64 stride 2, LeakyReLU. |
| $4 \times 4$ conv. 64 stride 2, LeakyReLU. |
| F.C. 2, $\pi(y) := y/\|y\|$. |

| Decoder |
| --- |
| Input $z \in \mathbb{R}^2$ s.t. $\|z\|_2 = 1$ |
| $4 \times 4$ deconv. 64, stride 2, ELU. |
| $4 \times 4$ deconv. 64, stride 2, ELU. |
| $4 \times 4$ deconv. 32, stride 2, ELU. |
| $4 \times 4$ deconv. 3,   stride 2, Sigmoid. |

# B  Proofs

We include the proofs for the propositions in the main text.

## B.1  Figure Eight Local Minimum

**Proposition B.1** *Assume that $f_{\phi(t)}$ undergoes continuous optimization and is thus continuous in $t$. Assume that $\pi \circ f_{\phi(t)} \circ g$ is injective for all $t$. The cyclic ordering induced on $k$ points by $f_{\phi(0)}$ is equal to that induced by $f_{\phi(1)}$. Thus a figure 8 embedding, which corresponds to cyclic order $(z_1, z_2, z_4, z_3) \bmod C_4$, cannot be transformed to a homeomorphic embedding, which has cyclic order $(z_1, z_2, z_3, z_4) \bmod C_4$ or $(z_4, z_3, z_2, z_1) \bmod C_4$.*

**Proof:** Since we assume $\pi \circ f_{\phi(t)} \circ g$ is injective for all $t$, the path $\mathbf{z}(t) = (\pi \circ f_{\phi(0)} \circ g(\theta_i))_{i=1}^4$ is inside the $k$-fold configuration space on $S^1$ defined $\mathrm{Conf}_k(S^1) = \{(z_1, \ldots, z_k) \in (S^1)^k : z_i \neq z_j$ for $i \neq j\}$. In order to prove the claim, we will show that the path-connected components of $\mathrm{Conf}_k(S^1)$ correspond to cyclic orderings of $(z_1, \ldots, z_k)$ and thus the start and end point of every path share a cyclic ordering.

Mapping $(z_1, \ldots, z_k) \mapsto (z_k, (z_k^{-1}z_1, \ldots, z_k^{-1}z_{k-1}))$ gives a homeomorphism $\mathrm{Conf}_k(S^1) \cong \mathrm{SO}(2) \times \mathrm{Conf}_{k-1}(S^1 \setminus \{1\}) \cong \mathrm{SO}(2) \times \mathrm{Conf}_{k-1}(\mathbb{R})$. Let $\tilde{z}_i = z_k^{-1}z_i$. Consider $D = \{(\tilde{z}_1, \ldots, \tilde{z}_{k-1}) : \tilde{z}_1 < \ldots < \tilde{z}_{k-1}\} \subset \mathrm{Conf}_{k-1}(\mathbb{R})$.

We can identify the connected components of $\mathrm{Conf}_{k-1}(\mathbb{R})$. The set $D$ is a fundamental domain for the action of the symmetric group $S_{k-1}$ on $\mathrm{Conf}_{k-1}(\mathbb{R})$. Thus $\mathrm{Conf}_{k-1}(\mathbb{R}) = \coprod_{\sigma \in S_k} \sigma(D)$ is a disjoint union. Linear interpolation shows $D$ is connected. The sets $D$ and $\sigma(D)$ for $\sigma \in S_k$ are not connected. Consider a path from $\mathbf{z} = (z_1, \ldots, z_k) \in D$ to $\sigma(\mathbf{z}) \in \sigma(D)$. The element $\sigma$ must reverse the order of at least two elements $z_j < z_i$. Thus the function $f(\mathbf{z}) = z_i - z_j$ must take the value 0 over the path by intermediate value theorem. Hence the path cannot be in $\mathrm{Conf}_{k-1}(\mathbb{R})$. Thus the connected components of $\pi_0(\mathrm{Conf}_{k-1}(\mathbb{R})) \cong S_{k-1}$.

Since $\mathrm{SO}(2)$ is connected, $\pi_0(\mathrm{Conf}_k(S^1)) \cong S_{k-1}$. That is each connected component of $\mathrm{Conf}_k(S^1)$ is labeled by an element of $S_{k-1}$ describing the ordering of $\tilde{z}_1, \ldots, \tilde{z}_{k-1}$ in $\mathbb{R}$. Each ordering of $(\tilde{z}_1, \ldots, \tilde{z}_{k-1})$ in turn corresponds to a different cyclic ordering of $z_1, \ldots, z_k$ in $S^1$, that is, a different element of $S_k/C_k$. Thus two $k$-point configurations are homotopic if and only if they have the same cyclic ordering. $\square$

---

[3] We used a higher number of random seeds than normal to account for the training instability.

## B.2 Degree Obstruction

**Proposition B.2** *The winding number of the initialized model and final model are equal* $w(\pi' \circ h_{\phi(1)} \circ g) = w(\pi' \circ h_{\phi(1)} \circ g)$.

**Proof:** The winding number of a map is a continuous function $t \mapsto w(\pi' \circ h_{\phi(t)} \circ g)$. Since the output space $\mathbb{Z}$ is discreet, the winding number must be constant in $t$. $\square$

## B.3 Magnitude Growth in $\mathcal{Y}$

*We assume that $dF$ is full rank*, which is a reasonable assumption for an overparameterized neural network. In that case $M = dF^T dF$ is a positive definite symmetric matrix and can be orthogonally diagonalized $M = Q\Lambda Q^T$ where $Q$ is orthogonal and

$$\Lambda = \begin{pmatrix} \lambda_1 & 0 \\ 0 & \lambda_2 \end{pmatrix}$$

and $\lambda_i > 0$. The maximum angle between $\boldsymbol{x} = \nabla_y \mathcal{L}$ and $M\boldsymbol{x} = \tilde{\nabla}_y \mathcal{L}$ can then computed in terms of the eigenvalues $\lambda_i$. This maximum is computed for the case of an $n \times n$ symmetric positive definite matrix here[4]. We include the proof for the $2 \times 2$ case we consider here for completeness.

**Lemma B.1** *The maximum angle between $x$ and $Mx$ for $x \in \mathbb{R}^2_{\neq 0}$ is*

$$\cos^{-1}\left( \frac{2\sqrt{\lambda_1\lambda_2}}{\lambda_1 + \lambda_2} \right).$$

**Proof:** The angle is maximized at the minimum value of

$$\frac{x^T M x}{\|x\|\|Mx\|}.$$

It suffices to consider $\|x\| = 1$. Substituting $M = Q\Lambda Q^T$ and $y = Qx$, we want to minimize

$$\frac{x^T Q^T \Lambda Q x}{x^T Q^T \Lambda^2 Q x} = \frac{y^T \Lambda y}{y^T \Lambda^2 y}$$

over all $\|y\| = 1$ since $\|Qx\| = \|x\| = 1$. Letting $a = y_1^2$ and noting $y_1^2 + y_2^2 = 1$, this is equal to minimizing

$$\frac{a\lambda_1 + (1-a)\lambda_2}{a\lambda_1^2 + (1-a)\lambda_2^2}$$

over $0 \le a \le 1$. Setting the derivative equal to 0 gives

$$\frac{(\lambda_1 - \lambda_2)^2(-\lambda_2 + a(\lambda_1 + \lambda_2))}{2\left(\lambda_2^2 + a(\lambda_1^2 - \lambda_2^2)\right)^{3/2}} = 0$$

and yields one critical value at $a = \lambda_2/(\lambda_1 + \lambda_2)$ corresponding to value $\frac{2\sqrt{\lambda_1\lambda_2}}{\lambda_1 + \lambda_2}$. This is the global minimum since the boundary values $a = 0$ and $a = 1$ correspond to maxima with value 1. $\square$

Thus the angle between $\nabla_y \mathcal{L}$ and $\tilde{\nabla}_y \mathcal{L}$ is bounded by $\theta = \cos^{-1}\left(2\sqrt{\lambda_1\lambda_2}/(\lambda_1 + \lambda_2)\right)$. Given that $\nabla_y \mathcal{L}$ is tangential to the circle, assuming for simplicity $\tilde{\nabla}_y \mathcal{L}$ has constant length $L$ and uniform distribution $[-\theta, \theta]$ in angle to $\nabla_y \mathcal{L}$, the norm of $y$ grows *in expectation*.

---

[4]karakusc (https://math.stackexchange.com/users/176950/karakusc), Maximum angle between a vector $x$ and its linear transformation $Ax$, URL (version: 2017-05-06): https://math.stackexchange.com/q/2266057

**Proof:** Without loss of generality, $y = (0, R)$ and $v = (L\cos t, L\sin t)$ where $|t| < \theta$. Then we evaluate

$$\mathbb{E}[\|y + v\|^2] = \frac{1}{2\theta}\int_{-\theta}^{\theta}\|(L\cos t, R + L\sin t)\|^2\|dt$$

$$= \frac{1}{2\theta}\int_{-\theta}^{\theta}(L^2\cos^2 t + R^2 + 2RL\sin t + L^2\sin^2 t)dt$$

$$= \frac{1}{2\theta}\int_{-\theta}^{\theta}(L^2 + R^2)dt$$

$$= L^2 + R^2.$$

by Pythagorean identity and the fact $\sin t$ is odd. $\qquad\square$

## C  Continuity Metric

For measuring continuity, we adopt a similar method as Falorsi et al. and evaluate continuity in terms of how the largest "jump" compare to others when walking a continuous path $m_i \in \mathcal{M}$ for $i = 1\cdots N$ pairwise close points. We define $q_i$ to be the ratio distances between the ground-truth and the learned representation

$$q_i = \frac{d_{\mathcal{M}}(\eta_\phi(m_i), \eta_\phi(m_{i+1}))}{d_{\mathcal{M}}(m_i, m_{i+1})}.$$

From the set $\{q_i\}_i$, we compute the continuity metric $L_{\text{cont}}$ as

$$L_{\text{cont}} = \frac{M}{P_\alpha}, \qquad M = \max_i q_i, \qquad P_\alpha = \alpha\text{-th percentile of } \{q_i\}_{i=1}^N. \qquad (6)$$

In our experiments, we set $\alpha = 90$.

There are two differences between how we evaluate continuity compared to [Falorsi et al., 2018]. First, we measure the continuity of $\psi_\phi$ rather than $f_\phi$, which we argue is more relevant. Second, in [Falorsi et al., 2018], the authors are mainly interested in verifying whether the encoder is discontinuous in the topological sense (which they verify by examining the inequality $M > \gamma P_\alpha$ for some $\gamma$). We on the other hand report continuity on the spectrum by computing the $\gamma$ that would make $\eta_\phi$ discontinuous. For evaluating homeomorphism, we conclude for an encoder to be homeomorphic if $L_{\text{cont}} < 10$ (empirically, we observed the mapping to appear smooth for a continuity score below this threshold).

# D Decoding From $\mathcal{Y}$

We consider the alternate strategy of removing the projection $\pi$ and adding a loss so that $y$ stays close to the desired manifold $\mathcal{M}$ in $\mathcal{Y}$. Winding number obstructions become far more prominent in this case. In Figure 7, we show the latent space for different random seeds in the teapot case when we train with such an objective.

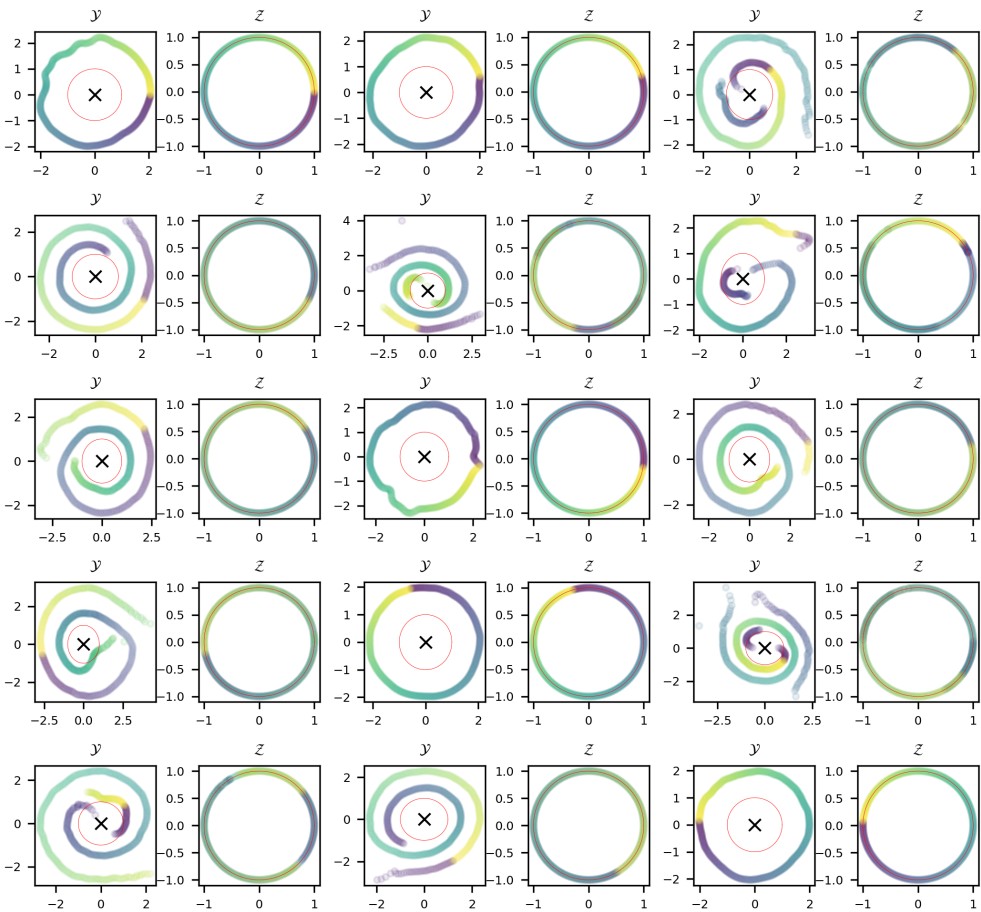

Figure 7: $\mathcal{Y}$ and $\mathcal{Z}$ space of Isom-AEs trained on teapots for 15 random seeds, where instead of decoding from $\mathcal{Z}$, we decode from $\mathcal{Y}$ with an additional soft regularization that constrain $y$ values to have unit length.

# E  Experiments: Additional Results

We report the full results of our SO(2) experiments in Table 3.

Table 3: Comparison of different VAEs trained on the various image datasets in terms of number of encoders with homeomorphic mappings (# H.), correct winding number (# W.), and correct crossing number (# C.) for 15 random seeds. We additionally report the error on continuity as well as negative Loglilkelihood (lower is better).

| L-shaped Tetrominoes | | | | | |
| --- | --- | --- | --- | --- | --- |
|  | # H. | # W. | # C. | Continuity | −Loglilkelihood |
| AE | 2/15 | 2/15 | 2/15 | $137.28 \pm 54.15$ | $9.66 \pm 5.63$ |
| VAE ($\beta = 1$) | 0/15 | 7/15 | 0/15 | $117.02 \pm 23.17$ | $7.26 \pm 6.99$ |
| VAE ($\beta = 4$) | 2/15 | 9/15 | 2/15 | $132.45 \pm 58.79$ | $\mathbf{2.67 \pm 0.30}$ |
| GF-VAE ($\beta = 1$) | 5/15 | 14/15 | – | $50.51 \pm 58.18$ | $2.71 \pm 1.67$ |
| GF-VAE ($\beta = 4$) | 9/15 | **15/15** | – | $24.04 \pm 29.14$ | $4.43 \pm 1.84$ |
| Action-GF-VAE ($\beta = 4$) | **10/15** | 14/15 | – | $\mathbf{19.35 \pm 23.37}$ | $4.43 \pm 1.67$ |

| Teapots | | | | | |
| --- | --- | --- | --- | --- | --- |
|  | # H. | # W. | # C. | Continuity | −Loglilkelihood. |
| AE | 0/15 | **15/15** | 8/15 | $173.37 \pm 21.99$ | $\mathbf{7.49 \pm 0.16}$ |
| VAE ($\beta = 1$) | 6/15 | **15/15** | 8/15 | $14.21 \pm 7.30$ | $7.61 \pm 0.07$ |
| VAE ($\beta = 4$) | **15/15** | **15/15** | **15/15** | $\mathbf{1.22 \pm 0.05}$ | $10.60 \pm 0.01$ |
| GF-VAE ($\beta = 1$) | 7/15 | **15/15** | – | $21.07 \pm 24.42$ | $8.48 \pm 0.38$ |
| GF-VAE ($\beta = 4$) | 13/15 | 14/15 | – | $8.16 \pm 17.89$ | $11.05 \pm 0.24$ |
| Action-GF-VAE ($\beta = 4$) | 13/15 | 12/15 | – | $3.14 \pm 1.74$ | $10.96 \pm 0.49$ |

| Airplanes | | | | | |
| --- | --- | --- | --- | --- | --- |
|  | # H. | # W. | # C. | Continuity | −Loglilkelihood |
| AE | 0/15 | 3/15 | 3/15 | $132.95 \pm 40.61$ | $11.55 \pm 2.19$ |
| VAE ($\beta = 1$) | 0/15 | 6/15 | 4/15 | $86.72 \pm 65.12$ | $12.61 \pm 3.41$ |
| VAE ($\beta = 4$) | 5/15 | 9/15 | **5/15** | $67.88 \pm 59.20$ | $12.53 \pm 1.77$ |
| GF-VAE ($\beta = 1$) | 0/15 | 11/15 | – | $63.94 \pm 34.43$ | $\mathbf{10.88 \pm 1.24}$ |
| GF-VAE ($\beta = 4$) | 7/15 | **13/15** | – | $\mathbf{27.70 \pm 29.23}$ | $13.17 \pm 1.64$ |
| Action-GF-VAE ($\beta = 4$) | **9/15** | 12/15 | – | $33.26 \pm 40.03$ | $12.27 \pm 1.79$ |

# F  Additional Figures

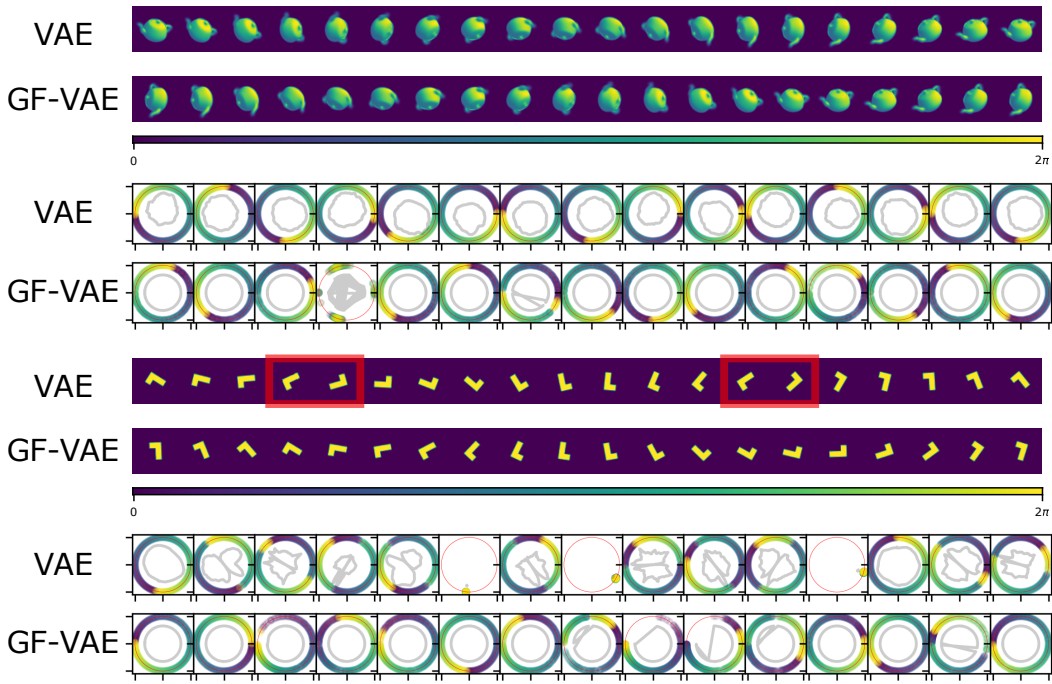

Figure 8: *Top*: Latent traversals in the decoder for a VAE and GF-VAE after training with $\beta = 4$. *Bottom:* The representations in latent space $\mathcal{Z}$ for VAEs and GF-VAEs initialized with 15 different random seeds. For VAEs, the gray line inside the circle show the (scaled-down) $y$-traversals. For GF-VAEs, we it is difficult to visualize $\mathcal{Y}$ space as it is high-dimensional space. Therefore, to inspect for obstructions, we show the traversal of the $z$ vectors instead.

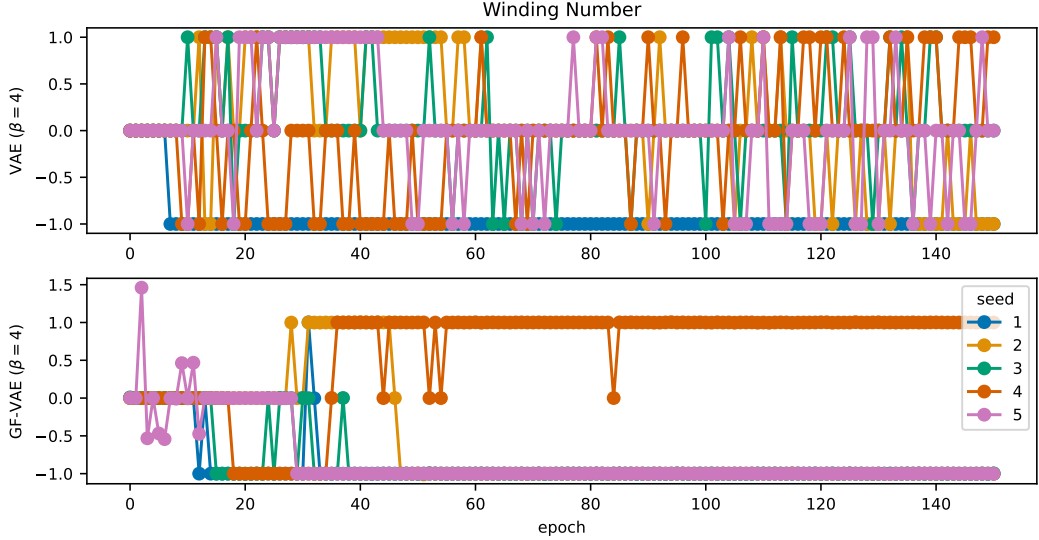

Figure 9: The winding numbers for VAE and GF-VAE trained with 5 different random seeds on the tetromino dataset as a function of epoch.

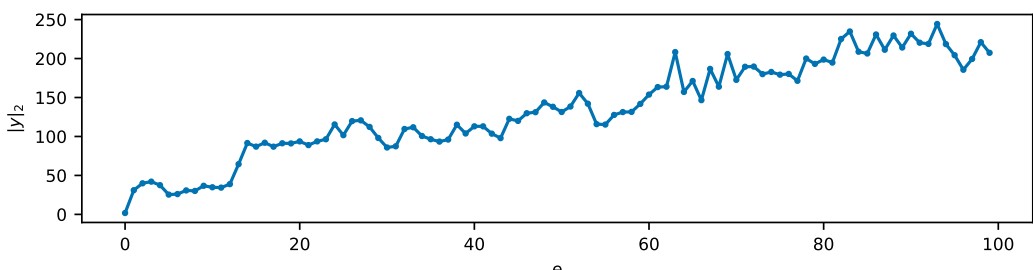

Figure 10: $\|y\|_2$ as a function of epoch for a standard autoencoder trained on teapots for seed 0.

