# OpenReview forum: "Topological Obstructions and How to Avoid Them"
_NeurIPS.cc/2023/Conference — NeurIPS 2023 poster_

### Official Review · Reviewer_WSwE · 2023-06-28

**Soundness:** 2 fair
**Presentation:** 3 good
**Contribution:** 2 fair
**Rating:** 6
**Confidence:** 3

**Summary:**

The authors investigate two types of topological obstructions that pose challenges for models aimed at learning a particularly structure in the embedding space. Specifically, the authors identify figure eight local minima and mismatches in winding numbers are two defects that make learning the right latent structure difficult. The authors propose a VAE method to improve the learning procedure in order to better reflect the underlying data’s structure. The authors run experiments on three shapes and measure continuity of the learned space in comparison to existing VAE models.


**Strengths:**

- Understanding challenges arising in the learning dynamics of models to ensure they appropriately reflect the underlying structure of data is an important topic.
- Authors acknowledge a reasonable set of limitations in the conclusion.
- It’s nice to see the authors applying tools from topology to examine how well the right structure can be learned by a model’s embedding space.


**Weaknesses:**

- The writing flow for Section 2 and 3 could use more work to improve clarity
	- Problem statement would benefit from an illustrative example of an application for f, h, and pi to help make the setup in line 61-64 more clear. I’d suggest the authors consider moving the running example earlier or using another application to motivate and illustrate this setup.
	- Figure 2 isn’t referenced at all in the running examples section and doesn’t appear in the text until Section 3 (line 107). I’d recommend moving the reference to Figure 2 earlier.
	- Figure 2: the figure would benefit from additional labels to indicate what each component is. For example, M as I understand is comprised of points for each degree of rotation for the object. The three right hand diagrams should be labeled clearly to indicate they are 3 runs (with different seeds).
- I found the empricial evidence, especially for the Torus experiment in 5.2 to be quite weak. For example, the authors base the results on only 5 runs grounding the conclusion on the fact that “VAE learns a homeomorphic mapping 0 times” while “GF-VAE learns the mapping 2 out of 5 times”. I don’t find this sufficiently convincing evidence as the authors test a single learning rate and setup across only 5 runs. I would recommend the authors run many more trials to prove this out.
- Unfair comparison to basleines: In the appendix, the full table shows Beta-VAE (with beta = 4) is quite an effective baseline, but this is absent from the main table 1 in the paper, which only compares against vanilla AE and VAE even though the GF-VAE model uses a beta term. This is not an apples-to-apples comparison to tease out the impact of the author’s proposed method. I suggest the authors include beta-VAEs in the main table and appropriately compare their method to the same beta-VAE baseline.
- I’m also surprised the authors did not include a comparison to other topological models that impose equivariance to Lie groups, but only to standard VAE models that do not impose group structure in the latent space.

Details
- typo in citation line 74
- typo line 227: “stand” → "standard”
- typo 239: extra space


**Questions:**

- Why does the setup include a projection function pi form y to Z? You cite Falorsi, but I couldn’t find intuition or motivation for this at all in the paper. Could you elaborate?
- How prevalent in practice is the figure eight local minima problem you identify in 3.1? This seems like a very specific topological artificat. Is it found real applications, or does it arise in training of deep networks?
- Could you explain the justification for the claim on line 178: “In order for f to be a homeomorphism, the winding number must be -1 or 1?”
- What exactly are the columns in table 1? The caption is quite sparse and doesn’t adequately describe how #H or Continuity are measured. The discussion of continuity seems to be relegated entirely to the appendix. At least a brief description with some intuition should be in the main text.
- What’s the rough computational cost of training with the additional normalizing flow regularization?
- How come the baseline “reg-y” model performs so poorly? I’m surprised that explicitly encouraging the latent space to be close to S1 doesn’t improve continuity on two of the three shapes.


**Limitations:**

Yes

---

> ### Author Rebuttal · Authors · 2023-08-09
>
> Dear Reviewer,
>
> Thank you for your kind review and constructive comments. We appreciate your thorough review and detailed questions! We are also delighted that you recognize the topic of our work as an important one. We believe we can address most of your concerns. We will respond to your questions and comments below:
>
> **Clarity of Section 2 and 3**:
>
> Thank you for your comments. We all make sure that figure 2 is referenced in the running example part as well as revising it to make it clear that the plots on the right belong to 3 different random seeds. We will also try to add a separate example at the beginning of Section 2, if there was sufficient space.
>
> **Tori Results**:
>
> Thank you for your constructive comment. The reviewer is completely correct. We were slightly short on time during the submission for the Tori experiment and therefore failed to run for 15 seeds or do any other hyperparameter tuning. We have run additional experiments since the deadline for the Tori experiments. GF-VAE (β = 6) converges to a homeomorphic mapping 9/15 times while 0/15 for VAE (β = 6). Hope this has addressed your concern.
>
> **Comparison against β-VAE**:
>
> Thank you for your constructive comment. We will make sure to add the β-VAE result in the main paper. As mentioned in our Appendix as well as [1], β was an important hyperparameter as it maximizes entropy which makes sure that all of the Lie group is covered and therefore improves continuity. However, as it is shown in the Appendix, you normally need both β and a GF-VAE architecture to learn a homeomorphic mapping (see β-VAE results for Tetrominoes and Airplanes).
>
> **Comparison against equivariant neural networks**:
>
> Thank you for your constructive comment. Equivariant neural networks achieve homeomorphism by definition so the reviewer is correct that if we have the prior knowledge of the Lie group *and* we can design an equivariance network for it, we should. However, as we are trying to build towards cases where we do not assume prior knowledge of the group or at least a design of an equivariant neural network for the group, here we consider a setting where no constrains are imposed on the neural network and only on the latent space. We hope that this and our general response and our response to Reviewer mNP4 have addressed your concern.
>
> **Why does $\pi$ exist?**
>
> Ordinary neural networks only map from Euclidean to Euclidean space. Therefore, if we want to map to a different space such as a Lie group, we have to add a separate mapping which is the role of $\pi$ in our design. For example, in the case of $S^1$, we can do this by either encoding to $\theta \in \mathbb{R}^1$ and then map to the circle $[\cos \theta, \sin \theta]$, or encode to $y \in \mathbb{R}^2$ first and then project to the circle $\pi(y) := y / \|y\|$. Hope this clarifies the role of $\pi$.
>
> **How prevalent is the figure-8 obstruction?**
>
> We mainly decided to analyze this obstruction because empirically this was the most common obstructions we faced in practice as you can see in the encodings in Figure 5. It is not unique in the sense that sometimes it manifests itself in the form of trefoil or figures with crossing numbers higher than 1 as well. However, we focus on figure 8 in our theory because it is the simplest case. Please see our general response for more details.
>
> **Why should the winding number be either 1 or -1 for a homeomorphic mapping?**
>
> Thank you for the question! The winding number can intuitively be explained by how many times do we ‘warp around‘ in the output space if we warp around once in the input space. Winding number 1 means as we go through $-\pi$ to $\pi$ (rotating counter-clockwise) in the input, the circle in the output space is also covered once in the same direction. Winding number -1 is basically the same thing except the circle in the output space is warped in the clockwise direction as we go through $-\pi$ to $\pi$ in the input space.
>
> **Details of our Evaluation Metrics**:
>
> Thank you for the question! We could have done a better job describing the details of our evaluation. The #H column shows the number of runs that have converged successfully to a homeomorphic mapping. Evaluating homeomorphism is difficult in general as it requires verifying continuity across the full domain. Here, we evaluated it based on two criteria: (1) if the continuity score is less than 8  (empirically, we observed the encoding to appear smooth below this threshold), and (2) winding number 1 or -1. The other column is continuity for which we describe the details of its calculation in the Appendix. This is done by taking an equidistant trajectory in the input space and keeping track of the pairwise distances in the output space. We then divide the maximum $q_i$ (where there was the most discontinuity) by the 90-th percentile $\{q_i\}$. We will provide more details of our valuation in the final manuscript.
>
> **Additional Computational cost of Flows**:
>
> This is a valid point and we will add it to our discussion of limitations in the paper. For the set of experiments in the paper, the additional computational cost was relatively small because in our experience, only a single flow layer was enough. Moreover, the flow is applied on the latent space which is very low dimensional therefore does not require a lot of extra computation. As an example, the training run time for a VAE on the Airplane dataset was 1 hour and 17 minutes while for GF-VAE it was 1 hour and 31 minutes.
>
> **$reg\text{-}y$ performance**:
>
> This was somewhat of a surprise to ourselves too. What in practice occurred when regularizing $y$-space was that if the mapping was already close to something homeomorphic, then it helped stabilize the convergence to a homeomorphic mapping. However, if the representation was stuck in a “figure 8” minima, then it made it even more difficult for the model to escape this local optimum as most of the arcs of “figure 8” had to be close to the circle and had no room to move.

---

> > ### Comment · Reviewer_WSwE · 2023-08-10
> >
> > Thank you for providing a detailed set of responses. In light of the additional runs for the Tori experiment, authors agreeing to move Beta-VAE to the main paper for direct comparison, and clarifications, I'm raising my score.
> >
> > I'd suggest including a few words about the authors' surprise regarding `reg-y` performance to aid readers and inspire future explorations.

---

### Official Review · Reviewer_1s1k · 2023-07-03

**Soundness:** 3 good
**Presentation:** 2 fair
**Contribution:** 2 fair
**Rating:** 5
**Confidence:** 3

**Summary:**

This paper theoretically and empirically characterize obstructions to training Homemorphic encoders with geometric latent spaces, such as local optima due to singularities or incorrect degree or winding numbers.

**Strengths:**

Originality: The paper is original in its approach to addressing topological obstructions in machine learning models. While previous work has explored the use of geometric inductive biases to improve interpretability and generalization, this paper is one of the first to systematically investigate the topological challenges of encoding to a specific geometric structure from the training perspective and propose a novel solution using adapted normalizing flows.

Quality: The paper is of fair quality, in terms of its theoretical rigor and its empirical evaluation. The authors provide a detailed analysis of the topological obstructions that can arise in encoders with geometric latent spaces, and their proposed GF-VAE model is well-motivated  to address these challenges.

Clarity: The paper isfairly written and organized. The authors provide ample background and motivation for their work, and their theoretical and empirical analyses are both presented in an accessible manner. The use of figures and examples throughout the paper helps to illustrate the authors' ideas and make the paper more understandable.

Significance: The authors' characterization of topological obstructions and their proposed GF-VAE model have the potential to improve the interpretability, generalization, and robustness of topological machine learning models. The paper is likely to inspire further research in this area and has the potential to lead to significant improvements in the performance and reliability of machine learning models.

**Weaknesses:**

While the paper is fairly written with novel contributions to the training of geometric deep learning, there are a few areas where it could be improved:

3. While the paper provides a clear and accessible presentation of the authors' ideas, some of themathematical concepts could be more clearly explained. For example, the motivation of homemorphic encoder and its applications in generative models should be discussed.     The concrete geometries considers in this paper are restrited to $S^1$, which is a commutative group. As th title is for general topological obstructios, the paper could benefit from providing a more general Lie group case.

2. The design of  GF-VAE is to apply flow model as a plug-in prior for the VAE, however combination of VAE and flow model is not new.  Therefore, there are likely other approaches and techniques that could be relevant to GF-VAE that are not discussed in detail.
 Second, The empirical evaluation of the proposed GF-VAE model is also limited to just two domains, and it is unclear how well the model would perform on other types of data. It would be helpful for the authors to provide additional experiments on a wider range of datasets to demonstrate the generalizability of their approach.

3. The paper could benefit from a more detailed discussion of the limitations of the proposed approach.

Typos: stand-> standard in line 227


**Questions:**

1. Why the author selected the figure eight and the degree cases to represent topological obstructions. Are there any principles for classifying topological obstructions?

2. What is the precise definition of the projection from $Y$ to $Z$ in formula (1) ?

3. Figure 2 doesn't contain the deconvolutional decoder $f^*$?

4. How is the flow $r$ realized (in line 227) for general lie groups?

5. The last paragraph of page 5 seemed to have provide a simpler solution to avoide topological obstructions than the  GF-VAE?

**Limitations:**

The paper could benefit from a more detailed discussion of the limitations of the proposed approach.

---

> ### Author Rebuttal · Authors · 2023-08-09
>
> Dear Reviewer,
>
> Thank you for your kind review. We are delighted that you recognized our theoretical and empirical contributions as significant and novel! We will respond to your questions and comments below:
>
>
> **Motivation for learning a homeomorphic representation**:
>
> Thank you for your comment. As stated in our introduction, we believe a reasonable notion of a good representation is a representation that reflects the underlying data structure, and we believe a sensible definition of this reflection is a representation that locally preserves the distances between the ground truth representations (and by extension, the data) which is described by the concept of homeomorphism. This should not only help us to perform better in downstream tasks, but also has the additional advantage of interpretability which is a desired property of any learned representation.
>
>
> **Does our theory apply to more general Lie Groups?**
>
> Thank you for your constructive comment. We believe there is a small misunderstanding. The concepts such as winding numbers and crossing numbers both can be defined for Lie groups of higher dimensions as well. We hope that our general response has addressed your concern.
>
>
> **Combination of VAEs and Flows**:
>
> Thank you for your constructive comment. Just to clarify, we apply to the flow to the inference model $q_\phi(z|x)$ and not the “prior” as the reviewer wrote. Nevertheless, we certainly did not mean to claim that we are the first paper that combined VAEs and Flows and will try to clarify this in the final manuscript. There are two main differences between our work and previous work that combined VAEs and Flows: (1) Previous work on Flow-VAEs only considered Euclidean latent spaces. To the best of our knowledge, we are the first work that employs a geometric latent space in this setting. This is mainly because defining flows on geometric spaces is a challenging task. (2) The motivation for applying flows to $q_\phi(z|x)$ in the prior work is to tighten the variational gap and therefore obtain a higher $\log p_\theta(x)$, while our motivation is to learn a homeomorphic mapping.  Does this address your concern?
>
> **Limited Scope of experiments**:
>
> We hope that our general response has addressed your concern. If the reviewer had a specific domain in mind, we would be happy to try.
>
> **Limitations**:
>
> Thank you for your constructive comment. We will expand on the limitations of our work mentioned in the conclusion.
>
> **Other types of topological obstructions**:
>
> That is a great question! Yes it would be possible to characterize different types of topological obstructions using tools from algebraic topology, namely homology and homotopy theory. Computing such topological invariants would give a classification of possible to obstructions to optimization, although many may be unlikely to occur in practice. The main reason we considered the figure-8 shape and the winding number mismatch was because empirically the combination of these two was the most common obstruction we faced in practice (Figure 5).
>
> **Definition of $\pi: \mathcal{Y} \rightarrow \mathcal{Z}$**:
>
> In the case of SO(2), $\pi(y)$ is just a projection on the circle: $\pi(y) := y / ||y||$. Thank you for your comment. We will clarify this in the final manuscript.
>
> **Decoder in Figure 2**:
>
> Our intention with this figure was to depict our encoder design as well as various types of potential representations we could learn from this design. As the decoder $f^*$ is just an ordinary neural network, we did to include it in that figure due to space constraints. We are happy to try to include it in Figure 2 if the reviewer feels it helps with clarity.
>
> **Normalizing flows for general Lie groups**:
>
> This is a great question! As mentioned in the related work Section, defining normalizing flows on geometric spaces is a field of itself. One general way of defining a flow on the Lie group is to define the flow on the Lie Algebra and use the exponential map to map it to the Lie group, which has been defined and discussed for most Lie groups. However, more care needs to be taken to account if we want to avoid having a discontinuous pdf as discussed in [1]. We refer the reviewer to [1,2] for more details on this topic.
>
> **Last paragraph on page 5**:
>
> If the reviewer is referring to the “directly decoding from $y \in \mathcal{Y}$ but push embeddings to the unit circle using the loss $| ||y|| − 1|$” we believe there has been a misunderstanding. As pointed out in the same paragraph and Figure 8 in the Appendix, optimizing this objective could result in the wrong winding number.
>
> **[References]**
>
> [1] Danilo Jimenez Rezende, George Papamakarios, Sébastien Racaniere, Michael Albergo, Gurtej Kan-war, Phiala Shanahan, and Kyle Cranmer. Normalizing flows on tori and spheres. In the International Conference on Machine Learning, pages 8083–8092. PMLR, 2020.
>
> [2] Luca Falorsi, Pim de Haan, Tim R Davidson, and Patrick Forré. Reparameterizing distributions on lie groups. In The 22nd International Conference on Artificial Intelligence and Statistics, pages 3244–3253. PMLR, 2019.

---

> > ### Comment · Reviewer_1s1k · 2023-08-18
> > **Response from the reviewer**
> >
> > I'm generally satisfied with the author's response. I will keep my 'marginal accept' score.

---

### Official Review · Reviewer_mNP4 · 2023-07-05

**Soundness:** 3 good
**Presentation:** 3 good
**Contribution:** 3 good
**Rating:** 5
**Confidence:** 4

**Summary:**

This paper explores the challenges of encoding data into geometric spaces and proposes a solution using Group-Flow Variational Autoencoders (GF-VAEs). The authors discuss how incorporating geometric inductive biases can improve interpretability and generalization but also present obstacles due to topological constraints. They identify two types of local optima that can arise: singularities and incorrect degree or winding number. To overcome these challenges, the authors propose GF-VAEs, which utilize normalizing flows to define multimodal distributions on geometric spaces. The paper characterizes topological defects in encoders, introduces evaluation criteria based on winding number, crossing number, and continuity, and demonstrates that GF-VAEs can escape local optima and achieve a more reliable convergence to a homeomorphic mapping. The main contributions of the paper include characterizing topological defects, proposing GF-VAEs as a solution, and empirically validating their effectiveness.

**Strengths:**

1. *Theoretical and Empirical Analysis*: The paper combines theoretical analysis with empirical evaluations to provide a comprehensive understanding of the challenges and solutions related to encoding data into geometric spaces. This approach strengthens the validity of the proposed methods and their practical implications.

2. *Identification of Obstructions*: The paper effectively identifies and characterizes topological defects that can occur in encoders mapping to geometric structures. By recognizing the specific challenges such as singularities and incorrect degree or winding number, the authors provide a clear understanding of the obstacles that need to be addressed.

3. *Proposal of GF-VAEs*: The introduction of Group-Flow Variational Autoencoders (GF-VAEs) as a solution to the identified obstructions is a significant contribution. The paper explains how GF-VAEs leverage normalizing flows to model complex multimodal distributions on Riemannian manifolds. This proposal offers a practical approach to circumvent local optima and achieve more reliable convergence.


**Weaknesses:**

1. *Idealized Assumptions*: The paper acknowledges that the theoretical analysis is limited by the idealized assumptions necessary to analyze the method using topological tools. These assumptions may not exactly match the real-world scenarios encountered in practice. This limitation undermines the direct applicability of the theoretical findings to real-world problems and raises questions about the generalizability of the proposed solutions.

2. *Limited Metrics for Higher Dimensions*: The metrics defined in the paper, such as winding number and crossing number, are primarily designed for lower-dimensional manifolds. It is mentioned that these metrics become harder to define and compute for higher-dimensional manifolds. This limitation restricts the applicability of the evaluation criteria to higher-dimensional geometric spaces, potentially limiting the scope of the proposed approach.

3. *Restricted Scope of Experiments*: While the paper presents empirical evaluations on two domains, it does not cover a wide range of datasets or scenarios. The limited scope of the experiments may not fully capture the diversity of real-world applications and datasets, leaving open questions about the performance and generalizability of the proposed GF-VAEs in different contexts.

4. *Lack of Comparative Analysis*: The paper lacks a comprehensive comparative analysis of the proposed GF-VAEs with existing methods. While the empirical evaluations demonstrate the effectiveness of GF-VAEs in escaping local optima and achieving better convergence, a thorough comparison with alternative approaches would provide a clearer understanding of the strengths and weaknesses of the proposed method in relation to existing state-of-the-art techniques.

**Questions:**

See the Section 'Weaknesses'.

**Limitations:**

The authors presented in the paper the most important limitations of the proposed approach.

---

> ### Author Rebuttal · Authors · 2023-08-09
>
> Dear Reviewer,
>
> Thank you for your kind review. We are delighted that you recognized our contributions as significant! We will respond to your questions and comments below:
>
> **Assumptions in Propositions 3.1 and 3.2**:
>
> The reviewer raises a valid point about the idealistic nature of our assumptions, as we acknowledge in our paper. However, undertaking theory work in a fully realistic setting is indeed challenging, particularly considering that, to the best of our knowledge, our work represents the first attempt to conduct theoretical analysis on the topic of topological obstructions. While we recognize the limitations of our assumptions, we would like to emphasize that these idealized conditions serve as a foundational step to understand the fundamental constraints and possibilities within the context of topological instructions. Our intention is to establish a theoretical framework that lays the groundwork for future research in this promising area. Moreover, it is essential to note that we adopt a perspective that interprets gradient descent as a discretization of a continuous path. Under this viewpoint, our theory highlights that even if continuous optimization were feasible, obtaining a homeomorphic mapping would not be possible. The presence of topological obstructions fundamentally means we have to break the continuity of the optimization path. This finding is an intriguing and novel result that showcases the inherent challenges in learning such mappings, and it holds significance irrespective of the idealized assumptions. As our field progresses, we anticipate that future works will build upon our theoretical framework to consider more realistic scenarios and explore techniques to address practical challenges.
>
>
> **Limited Metrics for Higher Dimensions**:
>
> We believe there is a small misunderstanding. The concepts such as winding numbers and crossing numbers both can be defined for Lie groups of higher dimensions as well. It is simply a matter of discretization of the latent space.  Though they become computationally more challenging to compute in higher dimensions. We will make sure to clarify this in the final manuscript. Please see our general response for more details.
>
> **Restricted Scope of Experiments**:
>
> Thank you for your constructive comment. We hope that our general response has addressed your concern. If the reviewer had a specific domain in mind, we would be happy to try it.
>
> **Lack of Competitive Analysis**:
>
> Thank you for your constructive comment. Due to the uniqueness of our setting, finding suitable prior works for a fair comparison presents challenges. Let us elaborate on the reasons behind our experimental choices and the considerations we took into account while designing our approach. In our work, we deliberately refrain from imposing constraints on the neural networks, such as equivariance, to maintain generality. This choice as well as the fully unsupervised nature of our setting aligns with our long-term goal of eventually moving towards scenarios where we do not assume prior knowledge of the correct Lie group. Thus, as a foundational step, we opt to utilize an ordinary encoder and solely impose constraints on the latent space itself. It is important to acknowledge that if we were to possess knowledge of the correct structure in advance, employing an equivariance neural network would be the correct choice. However, as mentioned in our related work, existing approaches either heavily rely on equivariance neural networks or assume additional information about the data, such as the presence of group elements 'g' in between pairs. These approaches, while useful in their respective contexts, deviate significantly from our fully unsupervised and unconstrained setting.  Also please note that the primary focus of our experiments was to investigate the hypothesis that multimodal distributions have a higher probability of learning a homeomorphic representation when compared to vanilla VAEs with a Lie group latent space. If the reviewer has any specific baseline that would like us to compare to, we would be happy to hear it.

---

### Official Review · Reviewer_qXsa · 2023-07-06

**Soundness:** 3 good
**Presentation:** 3 good
**Contribution:** 3 good
**Rating:** 6
**Confidence:** 1

**Summary:**

The paper addressed several topological obstructions that cannot be easily solved during the optimization of VAE. To solve this problem, the paper proposed to train a special NF to escape the defect, which the authors called GroupFlow. The paper then evaluated the proposed method on synthetic image datasets with different kinds of manifolds.

**Strengths:**

The paper is clearly written and the mathematical formation of the problem is precise.

The analysis on the obstructions during optimization clearly shows the invariants which lead to defects.

The proposed method naturally follows the defects and seems interesting to me.

The evaluation results indicate the proposed method achieves the best homeomorphism.

**Weaknesses:**

I am not an expert in topology, so please refer to other reviewers for the comments on theoretical analysis. However, it seems the authors only analyzed a very simple type of manifold with small freedom. It would be more interesting if there are results on more complex manifolds beyond rotation and colorization. There is a similar concern for experiments.

From Table 1 it seems the $\beta$ parameter has large impact on the homeomorphism result. There should be more (intuitive or theoretical) discussion on the relationship between $\beta$ (disentanglement) and homeomorphism. It is also important to demonstrate results from the standard $\beta$-VAE.


**Questions:**

Does your theory apply to more complicated manifolds?

Why does higher $\beta$ leads to better homeomorphism? How about results of the standard $\beta$-VAE?

**Limitations:**

The paper has adequate discussion on limitations.

---

> ### Author Rebuttal · Authors · 2023-08-09
>
> Dear Reviewer,
>
> Thank you for your kind review. We are delighted that you found our theory and proposed approach clear and interesting! We will respond to your questions and comments below:
>
> **Applicability to other Lie groups**:
>
> Thank you for your question! Yes, our theory does apply to other compact Lie groups as well. Please see our general response for more details.
>
> **Role of β**:
>
> The reviewer is absolutely correct that β indeed makes a difference. We show the results for β-VAE in the Appendix in Table 3. We believe the reason for its effectiveness is that because the prior $p(z)$ here is uniform, regularizing the KL terms corresponds to maximizing entropy which encourages the encodings to cover all of the latent Lie group (which is necessary for a homeomorphic mapping). As the results in Table 3 shows, sometimes increasing the β can be enough to learn a homeomorphic mapping (e.g. teapots). However, as we can see in the other cases, we generally need both a GF-VAE and a high β to achieve a homeomorphic mapping (e.g. Tetrominoes & Airplanes). We will make sure to add the results from β-VAE to the main paper as well.

---

> > ### Comment · Reviewer_qXsa · 2023-08-18
> > **Thanks for the reply**
> >
> > Thanks the authors for addressing my questions. The answers seem to be convincing. I will keep my score because I am not expert enough to evaluate the correctness and the impact of the theoretical (especially the topology) part of the paper. I also ask AC to give higher weights to the comments from other reviewers who are more expert in this topic.

---

### Author Rebuttal · Authors · 2023-08-09

We sincerely appreciate the valuable feedback provided by all the reviewers. Your constructive comments and efforts in evaluating our paper are highly appreciated. We are pleased to see that the general consensus is that our theoretical analysis of topological obstructions, along with the proposed GF-VAE, are significant and novel contributions to the field. There are some shared concerns among reviewers regarding the application of our theory and method to other domains and topological spaces, which we will address below before addressing individual comments:

**Applicability to other Lie groups and topological obstructions**:

All reviewers inquired about the applicability of our theoretical analysis beyond the specific case of $\mathrm{SO}(2)$ and “figure 8” shape. The wording in the conclusion might have made it seem the mathematical concepts we employ are limited to $\mathrm{SO}(2)$, but this is not the case. To clarify, the concepts of winding number and crossing number can be extended to other Lie groups and topological spaces of higher dimensions.  In higher dimensions, the winding number is known as the degree of a continuous mapping and can be computed over any compact manifold. Thus Prop 3.2 may be readily generalized. The crossing number can be generalized in different ways including the measure of the size of the self-intersection. For instance, we can consider a 2D sheet intersecting itself in a manner akin to a twisted ribbon.  Additionally, in the case of the figure-8 example, we note that other shapes with the incorrect crossing number, such as trefoil or quatrefoil, can also exist as a possible encoding and would be part of proposition 3.1. Computing these metrics for other topological spaces primarily involves discretization of the latent space, though we recognize that the computational challenges increase with higher dimensions. We apologize for any confusion caused by our phrasing, and we will rectify this in the final manuscript to make it clear that the defined metrics can be applied to various Lie groups and topological spaces.

**Limited Scope of Experiments**:

Reviewers mNP4, 1s1k, and WSwE pointed out that we are only considering Circles and Tori in our experiments.  In the field of equivariance/disentanglement, and generally extracting the right Lie group from data, $\mathrm{SO}(2)$ and $\mathrm{SO}(2) \times \mathrm{SO}(2)$ are the two of the most common Lie groups. We could consider adding translation to the features as well (e.g. $(R^2, +) \times \mathrm{SO}(2)$), but translation can be modeled with a standard Euclidean latent space so therefore does not lead to the topological obstructions discussed in the paper. The only common Lie group we did not consider in our experiments was the Lie group $\mathrm{SO}(3)$ which we will try our best to add to the camera ready. However, we would like to point out that besides homeomorphic-VAE, our work would be the only VAE-based work that would manage to learn a homeomorphic mapping from images to $\mathrm{SO}(3)$ in a fully unsupervised manner that doesn’t use an equivariance neural network. Moreover, even in the case of homeomorphic-VAE, it requires two additional regularizers as well as a hard tuned β-scheduler to have a single successful run. While we understand the reviewers' desire to see broader experiments, we feel that given the novelty of our theoretical and methodological contributions, the focus on circles and tori is sufficient for this paper. However, we assure the reviewers that we will explore the extension of our approach to other Lie groups in future work.

---

### Decision · Program_Chairs · 2023-09-21

**Decision:**

Accept (poster)

**Comment:**

This paper explores geometric inductive biases  and shows that it can improve interpretability and generalization but it is challenged by obstacles in the optimization due to topological constraints. Authors discuss  two types of local optimas that can arise: singularities and incorrect degree or winding number. The paper proposes GF-VAEs that uses  normalizing flows to define multimodal distributions on geometric spaces to alleviate these issues. The paper  introduces evaluation criteria based on winding number, crossing number, and continuity, and demonstrates that GF-VAEs  escapes local optima and converges to a homeomorphic mapping.

Reviewers agreed that the contribution of the paper is significant , although limited in scope in terms of the theory and the applications that consist only in two domains and lack of baseline comparisons. Authors clarified the applicability to other lie groups in the rebuttal and clarified and extended experiments on tori , and  promised to move the beta vae experiment from appendix to main, addressing overall reviewers concerns. Please take into account in the revision all reviewers comments and suggestions.